

# What caused the extreme CO concentrations during the 2017 high pollution episode in India?

Iris N. Dekker[1,2], Sander Houweling[1,3], Sudhanshu Pandey[1,2,3], Maarten Krol[1,2,4], Thomas Röckmann[2], Tobias Borsdorff[1], Jochen Landgraf[1], and Ilse Aben[1]

[1]SRON Netherlands Institute for Space Research, Utrecht, 3584 CC, the Netherlands
[2]Institute for Marine and Atmospheric Research Utrecht, Utrecht University, Utrecht, 3584 CA, the Netherlands
[3]Department of Earth Sciences, Vrije Universiteit Amsterdam, Amsterdam, 1081 HV, the Netherlands
[4]Department of Meteorology and Air Quality, Wageningen University and Research Centre, Wageningen, 6708 PB, the Netherlands

**Correspondence:** Iris N. Dekker (i.dekker@sron.nl)

**Abstract.** The TROPOspheric Monitoring Instrument (TROPOMI), launched 13 October 2017, measures carbon monoxide (CO) concentrations in the Earth's atmosphere since early November 2017. In the first measurements, TROPOMI was able to measure CO concentrations of the high pollution event in India of November 2017. In this paper we studied the extent of the pollution in India, comparing the TROPOMI CO with modelled data from the Weather Research and Forecast model (WRF)

to identify the most important sources contributing to the high pollution, both at ground-level and in the total column. We investigated the period between 11 and 19 November 2017. We found that residential and commercial combustion was a much more important source of CO pollution than the post-monsoon crop burning during this period, which is in contrast to what media suggested and some studies on aerosol emissions found. Also, the high pollution was not limited to Delhi and its direct neighbourhood but the accumulation of pollution extended over the whole Indo-Gangetic Plain (IGP) due to the unfavourable

weather conditions in combination with extensive emissions. From the TROPOMI data and WRF simulations, we observed a build-up of CO during 11-14 November and a decline in CO after the $15^{th}$ of November. The meteorological situation, characterized by low wind speeds and shallow atmospheric boundary layers, was most likely the primary explanation for the temporal accumulation and subsequent dispersion of regionally emitted CO in the atmosphere, emphasizing the important role of atmospheric dynamics. Due to its rapidly growing population and economy, India is expected to encounter similar pollution

events more often in future post-monsoon and winter seasons unless significant policy measures are taken to reduce residential and commercial emissions.

## 1 Introduction

During November 2017, India encountered an extreme pollution episode. Various ground-level measurement stations reported Air Quality Index (AQI, http://aqicn.org/) values of 999, i.e. far above the standard scale that is limited to 500. These high

AQIs were caused by high concentrations of several pollutants, but most importantly particulate matter ($PM_{10}$ and $PM_{2.5}$) with reported values of >700 $\mu$g/m$^3$ $PM_{2.5}$, and carbon monoxide with values of up to 10 mg/m$^3$. Most of this pollution was



found over the Indian part of the Indo-Gangetic Plain (hereafter called IGP), a highly populated region in the North of India near the Himalayas, including the cities of Delhi, Agra, Kanpur, Lucknow, Patna and Kolkata. Heavy air pollution is an annual recurring problem in this region, especially during the post-monsoon and winter months (Cusworth et al., 2018; Vadrevu et al., 2011; Girach and Nair, 2014). In the 2018 WHO list of most polluted cities in the world based on fine particulate matter

($PM_{2.5}$), based on 2016 data, nine out of ten cities were located in the IGP (Bhattacharya, 2016; WHO, 2018). At several ground-based measurement stations in the IGP maintained by the Central Pollution Control Board (CPCB, http://cpcb.nic.in/), carbon monoxide (CO) levels amply exceeded the world health organization guidelines ($100$ mg/m$^3$ for 15 minutes, $10$ mg/m$^3$ for 8 hour) during several days in November 2017 reaching values up to $400$ mg/m$^3$ ($1$ mg/m$^3$ is roughly equal to 870 ppb at ground level).

Several explanations have been proposed for the high pollution levels in this period, but the exact cause is still unclear. Agriculture is very important in the IGP, with the post-monsoon burning of crop residues taking place in October and November. In addition, post-monsoon meteorological conditions can lead to an accumulation of pollutants in Northern India (Liu et al., 2018). CO emissions from fires during this period are estimated, for example, by the Global Fire Assimilation System (GFAS) and the Global Fire Emission Database (GFED). However, there are strong signs that these datasets do not capture all of the

biomass burning emissions (Mota and Wooster, 2018; Cusworth et al., 2018; Huijnen et al., 2016). Also, the conversion of fire occurrence to CO emissions depends on factors such as combustion efficiency, biome type, and soil characteristics, which adds uncertainty to the emission estimates (Werf van der et al., 2010). Other anthropogenic sources of CO, e.g., traffic and heating systems are very high in the highly populated Indo-Gangetic plains, especially during the colder post-monsoon and winter months. Thus, apart from fire emissions from post-harvest burning practices, also these other anthropogenic sources might be

an important factor explaining the high CO pollution.

  Satellite data can play an important role to obtain more insight in the origin and extend of pollution, providing information on the distribution of pollutants over large regions on a daily basis. In October 2017 the TROPOspheric Monitoring Instrument (TROPOMI) was launched, measuring various trace gases, including CO, with unprecedented high spatial and temporal resolution (Landgraf et al., 2016). TROPOMI was still in its commissioning phase in November and algorithm tests and calibrations

were ongoing. Fortunately, the first calibration results were positive and proved the high quality of the measurement instrument and high signal to noise ratio (Borsdorff et al., 2018a, b), confirming the usefulness of the orbits of scientific data that were collected. TROPOMI observed very high column mixing ratios over the Northern part of India, in accordance with the ground-based data. The Copernicus Atmosphere Monitoring Service (CAMS, see section 2.2) data showed similar enhancements in CO columns during this period, further corroborating the TROPOMI retrieved CO variations (Borsdorff et al., 2018a).

In this study, we examine the high CO pollution episode of November 2017 in more detail. We do this by combining the daily CO observations over India from TROPOMI with ground-based measurements and simulated CO mixing ratios from the Weather Research and Forecast (WRF) model. We assess this according to our four objectives: (1) whether TROPOMI is in accordance with ground-based measurements and (2) how well WRF is able to reproduce these data. We investigate the dispersion of pollution over India to (3) shed more light on sources contributing most to the high pollution over the





Indo-Gangetic Plain (IGP) of India in support of future pollution mitigation efforts. With WRF we (4) also study the role of meteorology in the accumulation and spreading of CO.

The data and methods section describes the datasets that are used and the setup of the WRF model. In the results section, CO levels measured by TROPOMI over Southeast Asia and by ground-level pollution measurement stations are compared

with WRF data. The model is used also to attribute the high total column average mixing ratios over India to specific emission categories as presented in section 4. In this section also the role of meteorological conditions is discussed as well as the results of sensitivity tests on CO chemistry in the model.

## 2  Data and methods

### 2.1  TROPOMI

TROPOMI has a shortwave infrared spectrometer module, from which the total column average mixing ratio (XCO) is retrieved using the measured radiance around 2.3 $\mu$m. Due to its high spatial and temporal resolution, TROPOMI is able to observe global CO vertical columns on a daily basis (Landgraf et al., 2016).

We used data from 14 orbits of TROPOMI retrieved between 11 and 19 November 2017 that covered the Northern part of India. As in the study of Borsdorff et al. (2018a) on the first TROPOMI CO results, we used XCO values that were retrieved

using the operational algorithm SICOR (Landgraf et al., 2016). TROPOMI data were filtered for clear sky observations, and cloudy sky observations with a cloud top height < 5000 m and an aerosol optical thickness >0.5. Borsdorff et al. (2018c) found that including low-level cloud data increased the amount of available measurements, while hardly affecting the ability to measure relatively small scale sources by applying the SICOR algorithm to data from the SCanning Imaging Absorption SpectroMeter for Atmospheric CHartographY (SCIAMACHY).

We removed the two most westward pixels of every swath, which suffer from a not yet resolved performance issue (Borsdorff et al., 2018a). The first validation study showed that the TROPOMI data is in good agreement with CAMS data with a global mean difference of +3.2% and a Pearson correlation coefficient of 0.97 (Borsdorff et al., 2018b). Moreover, only a small mean bias of 6 ppb, with a standard deviation of 3.9 and 2.4 ppb for respectively clear and cloudy skies has been found compared to ground-based total column measurements of TCCON (Total Carbon Column Observing Network). The signal-to-noise ratio of

TROPOMI is high compared to previous satellite instruments retrieving CO (Borsdorff et al., 2018a).

The TROPOMI averaging kernel (AK) provides information on the vertical sensitivity of the satellite instrument for each single retrieved CO column (Borsdorff et al., 2014). The relationship between the reported CO vertical profile ($C_{retr}$ and the true CO profile ($C_{true}$) is given by Eq. 1. In this equation $C_{retr}$ is the retrieved CO profile and $C_{prior}$ the a priori CO profile. According to Borsdorff et al. (2014):

$$C_{retr} = C_{prior} + \boldsymbol{AK}(C_{true} - C_{prior}) \tag{1}$$



In this study, we compare the CO columns from TROPOMI, derived from $C_{retr}$, with the modelled columns from WRF. To make a fair comparison between the TROPOMI CO columns and the modeled CO columns, the AK has been applied in the same way to the modeled CO vertical profile (Eq. 1), by replacing $C_{true}$ with the modelled profiles.

## 2.2 CAMS

The Copernicus Atmosphere Monitoring Service (CAMS, https://atmosphere.copernicus.eu) provides data on air quality in 6-hourly time intervals at a global resolution of 0.25°x0.25°. The CAMS CO reanalysis product is derived from the output of a 4-dimensional variational (4D-Var) data assimilation system, based on ECMWF (European Centre for Medium range Weather Forecast) numerical weather prediction reanalysis data. It uses MACCity anthropogenic emissions, which combines information from the European Union MACC (Monitoring Atmospheric Composition and Climate) and CityZen (megacity Zoom for the Environment emission database, Granier et al., 2011) inventories. For biomass burning the GFAS fire emission inventory is used, which is based on MODIS fire counts and is provided at a 0.1°x0.1°resolution. The CAMS model is constrained by CO satellite observations from the Measurements of Pollution in the Troposphere (MOPITT) and the Infrared Atmospheric Sounding Interferometer (IASI) satellite instruments. Constraining the model with satellite observations provides a relatively good estimate of the actual XCO over the globe. Biases are found to be within ±10% with respect to satellite and TCCON observations according to the latest validation report (KNMI, 2018), with data delivery lagging behind real-time by about one week. In this research we used the CAMS CO reanalysis products at various pressure levels and the total column product (http://apps.ecmwf.int/datasets/data/cams-nrealtime/levtype=sfc/).

## 2.3 WRF

To model XCO and ground concentrations at high spatial resolution we used WRF version 3.8.1 (http://www.wrf-model.org/) with the Advanced Research WRF core (ARW). WRF is a numerical non-hydrostatic model developed at the National Centers for Environmental Prediction (NCEP). It has several choices of physical parameterizations, allowing application of the model to a large range of spatial scales (Grell et al., 2005). Our model domain of 2900 km by 2010 km is over the northern part of India and parts of Pakistan, Nepal, China and Bangladesh, including parts of the Himalaya mountain range (see Fig. 1a). Our model employed a 10x10 km$^2$ resolution and 29 vertical eta levels, and used the Mellor-Yamada-Janjic (MYJ) planetary boundary scheme (Janjic, 1994), the Unified Noah land surface model for surface physics (Ek et al., 2003; Tewari et al., 2004), and the Dudhia scheme (Dudhia, 1989) and the Rapid Radiative Transfer Method (RRTM) for short-wave and long-wave radiation (Mlawer et al., 1997). Cloud physics are solved with the Grell-Freitas cumulus physics ensemble scheme (Grell and Freitas, 2014).

Our boundary and input meteorological conditions, on 6-hourly basis, were based on ECMWF reanalysis data, similar to the CAMS model. WRF calculates its own meteorology in between these 6-hourly time steps and nudges towards the meteorological boundary conditions every 6 hours. The boundary conditions for CO were from the CAMS CO data on pressure levels, interpolated to the WRF model levels.





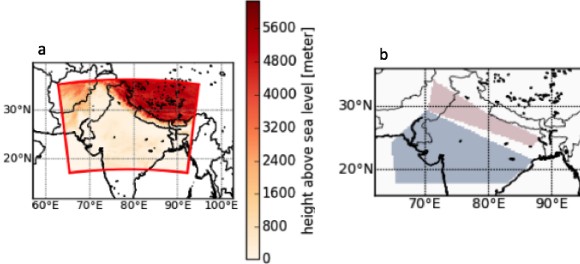

**Figure 1.** a) WRF domain over India, the colours depict the height above sea level (m), showing the Himalaya mountain range. (b) Within the WRF domain: the area for averaging over the IGP (pink) and for averaging over Non-IGP India (blue).

**Table 1.** Overview of the different tracers implemented in WRF.

| Tracer (short name) | Explanation | Source |
|---|---|---|
| en_prod | Energy production and distribution | MACCity |
| res_com | Residential and commercial combustion | MACCity |
| agr_waste | Agricultural waste burning | MACCity |
| ind_proc | Industrial processes and combustion | MACCity |
| agr_prod | Agricultural production | MACCity |
| solv_proc | Solvent production | MACCity |
| land_transp | Land transport | MACCity |
| mar_transp | Maritime transport | MACCity |
| waste_treat | Waste treatment and disposal | MACCity |
| COgfas | Biomass burning | GFAS |
| CObg | Boundary condition, referred to as *background* | CAMS |

Different CO emission inventories are available for Southern Asia. As in CAMS, we used MACCity anthropogenic CO emissions for the year 2017 at a resolution of 0.5°x0.5°(Lamarque et al., 2010). WRF-Chem simulations are performed for eight different CO tracers representing MACCity emission categories and one representing GFAS biomass burning (see Table 1). The MACCity database estimates worldwide monthly emission strengths for nine different emission categories. For biomass burn-

5    ing emissions, we used GFAS data with a resolution of 0.1°x0.1°(available for download from: http://apps.ecmwf.int/datasets/data/cams-gfas/). An additional tracer was used to account for CO transported from the CAMS derived boundary conditions: we refer to this CO tracer as background in this paper.





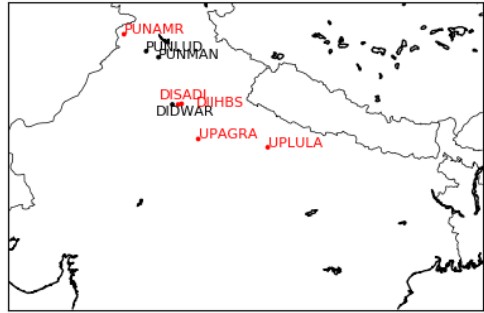

**Figure 2.** Locations of ground-level measurement stations, red dots and labels are inner city stations, black dots and labels outer city locations. PUN = Punjab: AMR = Amritsar, LUD = Ludhiana, MAN = Mandi Gobindargh; DI = Delhi: SADI = Sadipur, IBS = IHBAS, DWAR = Dwarka; UP = Uttar Pradesh: AGRA = Agra, LULA = Lucknow Lalbagh

Deposition and chemical production from Volatile Organic Compounds (VOCs) are not included in our base setup. The deposition process is slow compared to transport of CO out of the model domain, and direct CO sources over the highly populated (IGP) of Northern India are much larger than the indirect source from VOC oxidation.

However, in a sensitivity simulation (see Section 4.3) we accounted for the chemical reaction between the Hydroxyl radical

(OH) and CO using the JPL recommended temperature and pressure dependent reaction rate (Burkholder et al., 2015). Carbon monoxide production from the oxidation of methane and other VOCs are included in this simulation as well. In this chemistry simulation, we used the CO production from the TM5-4DVAR system (Krol et al., 2013) and the corresponding OH climatology based on Spivakovsky et al. (2000) and scaled by 0.92 (Huijnen et al., 2016, 2010; Krol et al., 2013)

## 2.4   Ground-level measurements

The central pollution control board (CPCB) of India measures the air quality at several stations in India (http://cpcb.nic.in/automatic-monitoring-data/). All the samples are taken at ground-level and are made available as fifteen minute averages. We only used stations here with CO measurements available between the $15^{th}$ of October and the $20^{th}$ of November. To obtain measurements representative of the urban background, we excluded stations near large roads showing large CO enhancements. This selection is needed for a meaningful comparison to WRF simulations at 10x10 km$^2$ using MACCity emissions at only

0.5°x0.5°resolution. In Fig. 2 all stations used for comparison with WRF are listed.

## 2.5   Comparing WRF with TROPOMI and ground-level measurements

As outlined before (section 2.1), the averaging kernel was applied to the WRF data using Eq. 1. Both WRF and TROPOMI data were averaged on a 0.25°x0.25°grid to make the comparison less sensitive to local outliers in the data. We also averaged over several days of data, concentrating on two periods: 11-14 November 2017, and 15-19 November 2017, in order to obtain




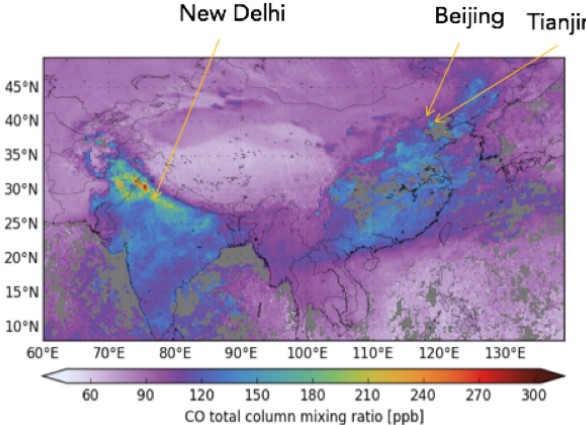

**Figure 3.** XCO over India and China as observed by TROPOMI, 13 November 2017, comparing New Delhi with large CO emitting cities in China, Beijing and Tianjin.

a gap free image of Northern India. We selected these two periods based on the patterns seen in TROPOMI data, the amount of data per day, and the weather conditions (see 4.3). In the first period, TROPOMI data show high CO pollution over the whole IGP. The second period shows lower XCO, due to changing weather conditions (see Section 4.2). In some comparisons, results are averaged over two regions of India: the IGP and the area south of the IGP: non-IGP, as defined in Fig. 1b.

We divided the ground-level measurement stations in India into two groups: one group consisted of stations directly in the city and the other group of stations was at the city edge in surrounding rural background regions (Fig. 2 respectively inner and outer city). This distinction was used to investigate differences in the source signature of CO inside and outside of cities.

## 3    Results

### 3.1    TROPOMI and CAMS over South-East Asia

In some of the first TROPOMI observations collected in the first half of November 2017, the northern part of India, more specifically, the IGP, stood out by its high XCO values (see Fig. 3). XCO values were even significantly higher over the IGP than over any region of South-East Asia, even higher than over China. This is remarkable, since in earlier studies, China used to be the most polluted region of the world (e.g., Baldasano et al., 2003; Kan et al., 2012). On the other hand, China has recently been active in reducing air polluting emissions, including CO (Zheng et al., 2018), while in India, emissions continued
to increase over the past years (Krotkov et al., 2016).

It was estimated that China reduced its CO emissions by 23% between 2013 and 2017 (Zheng et al., 2018). India only took its first steps to improve the air quality in December 2017 by implementing the National Clean Air Program (NCAP), i.e., after the high pollution event studied in this paper. This makes it plausible that the New Delhi region was more polluted in this



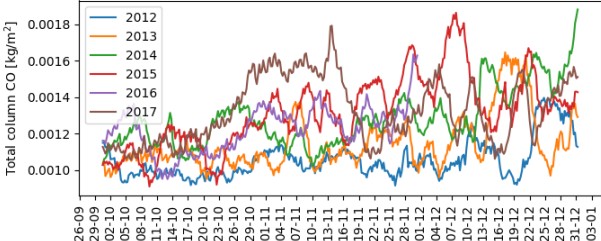

**Figure 4.** CAMS CO columns (kg/m$^2$) from October to December (2012 - 2017) averaged over the IGP domain (see Fig. 1)

period than any region over China. NCAP does not yet include strict targets for emission reductions (Ganguly, 2018; Ministry of Environment, Forest and Climate Change (MoEF&CC), 2018).

However, how unique were these high CO values during this time of the year in India over the IGP? We analysed the last four years of Copernicus Assimilation and Monitoring Service (CAMS) data to investigate this. These results confirm that the

high pollution episode of November 2017 was exceptionally long with more than two weeks of CO column amounts exceeding 0.0015 mg/m$^2$. However, according to the CAMS model, CO columns reached short-term values that were higher during December 2014 and 2015 than in November 2017 (Fig. 4). The high CO columns of November 2017 are therefore not unique for this part of India. However, high pollution episodes during the post-monsoon period occur more frequently in recent years (Fig. 4). As long as emissions are not reduced, India will probably encounter such events more often in future post-monsoon

and winter seasons.

## 3.2 Comparing WRF to TROPOMI

We compared our WRF results with the TROPOMI data, and found that WRF could reasonably well reproduce the high XCO values spread over the whole IGP during 11-14 November 2017 and the lower XCO values during 15-19 November. Fig. 5 shows that both modelled and TROPOMI retrieved XCO are very high in the north-west of India (Fig. 5, panels a, c, and e). The

highest spatial correlation between WRF (including standard GFAS emissions) and TROPOMI is found on November 11, 12, and 13 (r=0.87, 0.88, 0.88 respectively). On November 14 and 15 poorer correlations of 0.78 and 0.76 are found, respectively. The model captures the transition between the two periods slightly different from the observations, with TROPOMI showing ventilation of CO to the south-east earlier than WRF. The spatial correlation went up again to 0.81 for 16 and 17 November and 0.85 for 19 November. The WRF simulation underestimates XCO during the 11-14 November period over the IGP (Fig.

5a). Adding either 20% extra MACCity emissions or adding substantial amounts of GFAS fire emissions: between 500% (Fig. 5e) and 1000% of regular GFAS emission, gave XCO values that are more similar to the TROPOMI values without notable changes in the spatial patterns over India. In all cases, WRF overestimated the CO levels at the border of Pakistan south of the IGP. For 15-19 November, however, the simulations with MACCity and standard GFAS emissions already overestimate the XCO measured by TROPOMI (Fig. 5b). This might have to do with a deviation in the changing meteorological conditions in

WRF. Not including the atmospheric chemistry is probably only playing a minor role (see section 4.3).





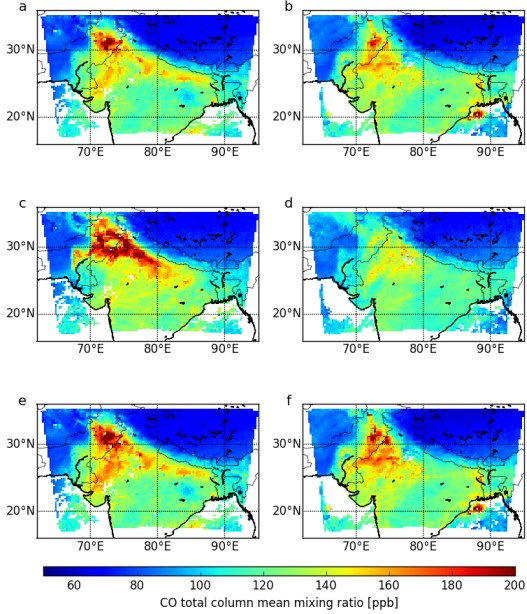

**Figure 5.** (a.) WRF simulated total columns with MACCity emissions, with 1x GFAS emissions, 11-14 Nov, (b.) same as a., but for 15-19 Nov, (c.) TROPOMI total columns 11-14 November, (d.) same as c. but for 15-19 November (e.) WRF simulated total columns with MACCity emissions including 5x GFAS fire emissions 11-14 November, (f.) same as e. but for 15-19 November.

### 3.3 CO columns over and outside of the Indo-Gangetic Plain

The XCO levels measured by TROPOMI and modelled by WRF are clearly enhanced during 11-14 of November over the IGP compared to more southerly regions of India (Non-IGP, Fig. 6). The IGP CO total columns are on average 30 ppb higher than then over non-IGP regions (see Fig. 1b for areas of IGP and non-IGP). When we average over 15-19 November, this difference
5  between the IGP and the non-IGP mostly disappears; the column average XCO over the Indo-Gangetic Plains is now lowered from 162 to 129 ppb for TROPOMI and from 152 ppb to 124 ppb for WRF, while the non-IGP XCO only slightly decreased for TROPOMI (124 to 118 ppb) and remained nearly equal at 129 ppb for WRF. A WRF simulation based on MACCity without GFAS (green bars), shows the same XCO pattern. Since the emissions of MACCity are not changing day-by-day, the difference between the periods is solely caused by different meteorological conditions (see also section 4.2).

10  ### 3.4 Comparing WRF to ground-level measurements

The CO concentrations at all ground-level measurement stations that are used are enhanced between the $3^{rd}$ of November and the $13^{th}$ of November, compared to earlier measurements. The CO diurnal cycles of the observations and the simulation show clear night-time accumulation in the stable nocturnal boundary layer, which vanishes during day-time with increasing





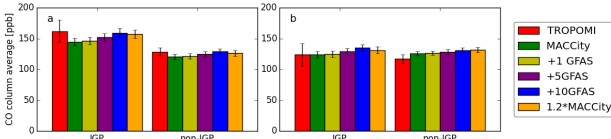

**Figure 6.** CO averaged over Indo-Gangetic Plains and non-IGP area. Panel (a.): for the period 11-14 Nov, panel (b.): for 15-19 November. The $1\sigma$ error bars denote the spread over the different days in the averaging period.

boundary layer depth. The CO concentrations generally reach lower levels after the $16^{th}$ of November (Fig. 7). The WRF model largely follows the CO enhancement and reduction pattern, although the diurnal cycle seems delayed by 3 hours compared to the ground-based measurements. This might be due to the hourly time profiles that were used for the emissions, which were derived for Europe (van der Gon et al., 2011), but do account for the local time shift. In Fig. 7, we zoom in on November 11-20,

the days for which also TROPOMI data are available. Averaged time series are shown of the measurements collected at stations in the provinces of Delhi, Punjab, and Uttar Pradesh and the corresponding averaged WRF concentrations. The stations inside the cities (Fig. 7a) show a clear reduction in mixing ratio during the latter half of this period (1050 ppb, 15-19 November), compared to the first half (1700 ppb, 11-14 Nov). The observed reduction, which we observed also in TROPOMI XCO, is reproduced by WRF (1400 to 880 ppb). At locations outside cities (Fig. 7b) this pattern is less pronounced, both in WRF and

in the measurements. WRF largely follows the measured CO mixing ratios, but slightly underestimates the CO values after the $16^{th}$ of November. WRF shows enhanced XCO during the $15^{th}$ and $16^{th}$ of November, which is not observed.

To further investigate the origin of the XCO variations, the contribution of different emission categories in WRF is shown in Fig. 8. We show here the inner-city stations, as these are the areas were most people live, but the picture is not very different for outer-city stations (see also Table 2, section 4.1). As can be seen, the surface concentrations are much less sensitive to

the background CO (black) compared to the total column mixing ratios. On all days the category *residential and commercial combustion* contributes most to the total CO concentration (on average 67% for ground-level and 35% for the total column including the background). Other large contributors are *industrial processes and combustion* and *traffic*. Surprisingly, we find a rather small contribution from fires to the total mixing ratio of 1-2% in our simulation with MACCity and standard GFAS emissions (see section 4.1, Table 2). Even with strongly enhanced GFAS emissions, the contribution remains on average

within 20%. The larger XCO measured at inner-city stations -compared to the stations outside the cities- also point to large contributions from urban emissions.

## 4  Discussion

We found XCO values of over 200 ppb in substantial parts of Northern India, in both the TROPOMI and model simulations. From the satellite data and total column WRF mixing ratios, it is clear that CO is not only enhanced directly around Delhi, but

over the whole Indo-Gangetic Plain, with very high values west of Delhi.



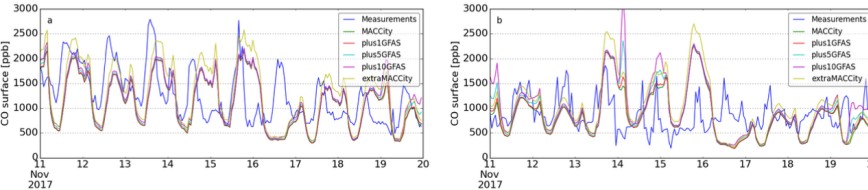

**Figure 7.** (a.)Ground-level measurements and WRF model averaged CO over *inner-city* stations (mg/m$^3$, see Fig. 2), time is in UTC (b.) Ground-level measurements and WRF model averaged over stations outside cities (mg/m$^3$, see Fig. 2)

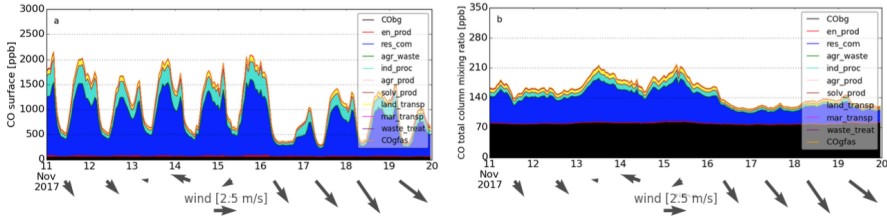

**Figure 8.** Average inner-city source contribution (a.): ground-level, time is in UTC (b.): inner-city total column mixing ratio. Below the mixing ratios, the 10-meter wind speed and direction are depicted, longer arrows are higher windspeeds (see legend)

Some ground stations were excluded because of the proximity to large CO sources, leading to very high observed CO concentrations. At the station of Harayana Rohtak, north-west of Delhi, the measured CO concentration was on average of 20 mg/m$^3$ from 11-14 November, but a value as high as 402 mg/m$^3$ was reached during a short peak. Our model could not resolve these high CO values, and also other nearby measurement stations did not record these high values. They are therefore very

likely caused by a local source that cannot be reproduced at the resolution of WRF. For this reason, also several other locations were excluded, where short-term CO peaks of several tens of milligrams per cubic meter were measured (stations: Bihar Gaya, Bihar Patna, Madya Pradesh Dewas, Madya Pradesh Ujjain, Rajasthan Jaipur Adarsh Nagar, West Bengal Durgapur). Although these measured enhancements are likely traffic related, it is well possible that some of these enhancements are caused by local biomass burning. Overall, however, our results point to a relatively small role of biomass burning in the enhanced CO

concentrations over the largest part of the IGP, as shown in Fig. 8 for inner-city stations. A similar contribution of biomass burning is found out of the city (not shown).

## 4.1    Contribution of different sources

According to the tracer simulations in our model that are based on MACCity and GFAS emissions, more than 50% of the CO near ground-level in November 2017 was caused by residential and commercial combustion. Other main contributors are

industrial combustion and traffic. For the total column mixing ratios the background CO, entering from the boundaries of the domain, was clearly more important and responsible for around a 30%-50% of the total column mixing ratio (see Fig. 4).



**Table 2.** Contribution of standard GFAS biomass burning to the total CO levels per region at ground level (GL) and in the total column (TC) between 11-19 November 2017.

| Period | | | Delhi GL | Delhi TC | Punjab GL | Punjab TC | Uttar Pradesh GL | Uttar Pradesh TC | Inner city GL | Outer city GL |
|---|---|---|---|---|---|---|---|---|---|---|
| 11-19 Nov | | maximum | 12% | 11% | 23% | 10% | 3% | 3% | 14% | 23% |
| | | average | 1% | 1% | 2% | 1% | 1% | 1% | 1% | 2% |
| 1 Oct - 19 Nov | | maximum | 17% | 15% | 44% | 16% | 15% | 11% | 40% | 44% |
| | | average | 2% | 2% | 6% | 1% | 1% | 1% | 2% | 5% |

The background CO is however rather constant, and day-by-day variations in XCO are caused by residential and commercial combustion, similar to what we observed at ground-level.

At the measurements stations that we considered, except for Punjab at ground-level, only a minor contribution from fire was found of 1% to 2%, both for total column and ground-level CO (see Table 2). At ground-level in Punjab the average and

maximum contributions were 6% and 44%, respectively over the whole modelled period of 1 October to 19 November. In the 11-19 November period, the maximum contribution of biomass burning to the ground-level contribution there was 23%, with an average of 2% (see Table 2).

There are strong indications that GFAS might severely underestimate the fire emissions (Mota and Wooster, 2018). Cusworth et al. (2018) concluded in their recent paper on biomass burning in India that the resolution of the MODIS satellite instrument,

on which GFAS fire emissions are partly based, misses many small fires. In addition, thick smoke from fires might lead to an underestimation of fire emissions from GFAS, as MODIS might identify these as clouds, as was found in a recent study over Indonesia (Huijnen et al., 2016). The results of increasing the fire emissions by a factor 5 - 10 in WRF are shown in Figs. 5 and 6. Adding biomass-burning emissions in the WRF simulation does not lead to a higher spatial correlation between WRF and TROPOMI but CO levels get closer to TROPOMI values in the 11-14 November period, so it might be that the GFAS

fire emissions were indeed underestimated in this period. However, the mixing ratios during the 15-19 November period are overestimated with respect to TROPOMI when higher GFAS emissions are assumed (see Figs 5,6). Alternatively, MACCity already explains a very large part of the observed CO levels, and increasing the MACCity emissions by 20% gives rather comparable results to increasing the GFAS fire emissions by a factor 5-10 for the total columns (Fig. 6). Biomass burning would still be a minor contributor to the average CO levels even if the emissions are enhanced by these factors.

For Delhi and Uttar Pradesh the contributions of fire emissions to the total CO levels are on average minor, even if the GFAS emissions are increased. For Punjab, we find that fire emissions might have contributed significantly to the ground-level concentrations for a few days in the 1 October to 19 November period. From the total column, and other stations, however, we



conclude that MACCity already explains a very large part of the observed XCO and ground-based CO levels, and fire emissions can only have played a very minor role.

In this paper, however, we assume that the emissions of MACCity do not grossly overestimate CO emissions over the IGP. Compared to TROPOMI and the amount of emissions that might come from fires based on GFAS and GFED, this assumption seems legitimate. When comparing the total emissions of MACCity to the EdgarV4.3.1 emission database of the most recent year 2010, however, EdgarV4.3.1 gives for November a circa 20% lower emission estimate, when taking emission factors of van der Gon et al. (2011) to convert from yearly emission to monthly emission. This gives more space to add extra emissions of GFAS, although we should keep in mind that in that case we have to increase the emissions of GFAS even more radically, in the order of >30x the original emissions.

## 4.2 Meteorological conditions

In general, the post-monsoon and winter season are the seasons with the worst air quality in the IGP. The photochemical loss is low and other meteorological variables, e.g., the absence of rain and low wind speeds, contribute to high levels of pollution.

For November 2017 we identified meteorological conditions as the most important reason why the CO mixing ratios at ground-level and in the total column increased as observed. Although not extreme, the meteorological conditions were favourable for the accumulation of air pollutants. The wind speeds near the surface were low for several weeks: < 2.5 m/s at 10 meter height, limiting the advection of CO away from the sources (Fig. 9). The temperatures were relatively low, decreasing from 22°C to 16°C from 1 Nov to 19 Nov, thus limiting vertical convection. The planetary boundary layer heights were low with daily averages between 350 and 580 m, diagnosed from WRF, while the air pressure (around 990 hPa) and relative humidity (up to 70%) were relatively high (Fig. 9). The most important changes that we found in meteorological parameters around the $15^{th}$ of November when the CO concentrations started decreasing, are in the wind speed, the wind direction, the relative humidity and the boundary layer height. The wind speeds clearly increased after the $15^{th}$ of November and the wind direction changed from a north-westerly direction to easterly winds in this period of the highest CO concentrations and the start of the ventilation. The relative humidity (RH) went up from 55% to 70% on November 15 and decreased afterwards to 45%. The boundary layer was highest on the $18^{th}$ of November (580 meter, see Fig. 9) but we found that more to the north-west of the IGP, closer to the Himalayas, boundary layer heights were also exceeding the height of this mountain ridge on November 14 and 15 (not shown). The highest CO values around Delhi were found during 13-16 November, so just before the winds were turning and increasing. In our WRF simulation, the most important contributors to the decrease in CO were both the ventilation of the IGP with clean air from the Himalayas followed by advection of the pollution to the south-east, which took place over all days after 15 November, and the outflow of CO towards the Nord-West, around the Himalayas, in the upper troposphere. We could clearly observe this outflow of CO in the upper layers of WRF on November 14 and 15 and it was also visible in the TROPOMI measurements on the same days (see also: Borsdorff et al., 2018a). The emissions of MACCity that went into the WRF simulation with only MACCity emissions were the same every day of November, which means that the increase and decrease in CO levels in the MACCity-only run (Figs. 6, 7) were due to the meteorological conditions.





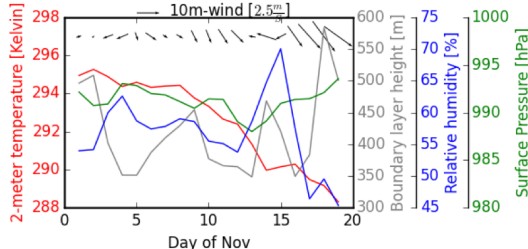

**Figure 9.** WRF 24-hour average meteorology over an area of 70x40 km$^2$ around Delhi and Agra in November 2017: 2-meter temperature (red, left axis), Boundary layer height (gray, right axis), relative humidity (blue, right axis), Surface pressure (green, right axis) and 10m wind speed and direction (black arrows, key gives length of 2.5 m/s wind). The transition from high to low CO levels is taking place around the $15^{th}$ of November.

Meteorological variability in relation with air pollution was studied before by Verma et al. (2017), who focused on Agra, Uttar Pradesh, 100 km South East of New Delhi. The meteorological conditions they reported for November 2015 were similar to what we found for November 2017. However, the RH peak of 70% was in the upper 25% of their measurements for November, and the boundary layer was shallower than the lowest level of 450 meter they found in their study. We should remark

here that the meteorology in our study is based on WRF model simulations, while in Verma et al. (2017), the meteorology came from automatic weather stations, except for the boundary layer height, which was obtained from the MERRA (Modern Era Retrospective-analysis for Research and Applications) reanalysis. Just looking at the data that we have, however, we can conclude that compared to the 2015 study, circumstances for CO accumulations were slightly more favourable for accumulation in November 2017. In our study, as in Verma et al. (2017), stagnant weather conditions are an important cause of the high

pollution levels.

### 4.3 The effect of chemistry and VOC emissions

In the base setup of WRF we did not include indirect CO sources from e.g., VOC oxidation in the analysis. Also, we did not include the OH oxidation of CO, and the only sink of CO was outflow at the lateral boundaries of the model. To assess the importance of VOC oxidation to the XCO and ground-level concentration, we performed a few sensitivity simulations,

including indirect CO sources from VOCs and methane and the oxidation of CO by OH. The most important differences were found over the region directly around New Delhi. XCO increased by up to 4% due to the oxidation of VOCs and methane. Oxidation of CO by OH lead to a decrease of up to 7% and, combining both mechanisms, a net decrease in XCO of up to 4% was found in the New Delhi region. In the non-IGP area, the effects of VOC oxidation, OH removal, and the two combined were respectively +2%, -5% and -2% on the XCO. Over the Indo-Gangetic Plain, effects are somewhere in-between. Patterns of

XCO enhancement over India were hardly affected by VOCs and OH oxidation. Compared to the uncertainty in the emissions, we consider these model simplification unimportant and therefore justified for the goals of this study.



## 5    Summary and conclusions

TROPOMI showed very high levels of XCO (>280 ppb) over Northern India during the high pollution event in India in November 2017. TROPOMI captured the spatial pattern of the pollution, covering not only Delhi, but rather the whole IGP. November is in the post-monsoon crop burning season, and media and scientific papers pointed to emissions from crop residue

burning as the main reason for the high pollution levels over the IGP (Jha, 2017; Vadrevu et al., 2011; Liu et al., 2018; Cusworth et al., 2018).

In this study, we analysed two consecutive periods in November: 11-14 November with the highest CO levels and 15-19 November, when CO levels decreased. High CO levels and a subsequent drop in CO were observed by TROPOMI, in ground-level measurements, and in our WRF simulations. The meteorological situation, characterized by low wind speeds and shallow

atmospheric boundary layers, was most likely the primary explanation for the temporal accumulation of regionally emitted CO in the atmosphere. The increase in wind speed and change of wind direction around 14 November led to the subsequent dispersion. The dominant role of meteorology, rather than emission variations, is supported by the fact that the WRF simulations that used constant emissions during the whole period, showed a similar temporal dependence, including decreasing CO levels after the $15^{th}$ of November.

After analysing the contribution of specific emission sectors to the simulated and observed CO levels over India, we conclude that residential and commercial combustion explain the largest fraction of the high CO pollution over the IGP. Biomass burning only plays a minor role in the CO enhancement: on average 1-2% at ground-level, and only 1% to the total column pollution level. In earlier studies, it was found that the GFAS biomass burning data, used in our analyses, likely underestimate the actual emissions of CO (Mota and Wooster, 2018; Cusworth et al., 2018; Huijnen et al., 2016). The comparison of TROPOMI data

with our WRF simulations, based on MACCity and GFAS data, confirms that CO emissions are underestimated in the 11-14 November period. The difference could be accounted for by increasing the GFAS emissions to 500%-1000% of the value in GFAS, a rather large increase compared to the findings of last named studies. In that case, the contribution of biomass burning to the observed pollution levels becomes more important: in the order of 5%-20%, but it would still remain smaller than the contribution of urban CO emissions. Therefore, unless urban MACCity emissions are largely overestimated and GFAS

emissions are underestimated even more, which we consider a less likely scenario, the contribution of urban CO emissions is the most important contributor to the CO pollution inside and out of the cities. These findings have important implications for emission mitigation efforts to avoid extreme pollution levels over the IGP during the post-monsoon period.

Our results have implications for ongoing winter time pollution mitigation efforts in India. Meteorology is found to be key driver of the extreme pollution episodes, which cannot be altered cheaply with the current state of geoengineering technologies.

Hence, to mitigate the pollution, reducing the largest CO emission sources (residential and commercial combustion) remains the best solution short-term and long-term.

*Data availability.*  Data used in this study can be found under ftp://ftp.sron.nl/open-access-data/



*Author contributions.* Iris Dekker, Sander Houweling, Maarten Krol, Thomas Röckmann, Sudhanshu Pandey and Ilse Aben were active in the conceptualization of the study, and designed the methodology. Iris Dekker performed the model simulations, carried out data analysis, and wrote the manuscript. Sander Houweling mentored Iris Dekker. Sudhanshu Pandey, Tobias Borsdorff and Jochen Landgraf helped with the resources: Sudhanshu Pandey with the acquisition of the Indian surface measurements and Tobias Borsdorff and Jochen Landgraf provided the TROPOMI data. All authors contributed to editing the manuscript.

*Competing interests.* The authors declare that they have no conflict of interest.

*Acknowledgements.* We would like to thank Antje Inness (ECMWF) for helping us with the analysis by providing CAMS data and explanation of the data. The GFAS data were generated using Copernicus Atmosphere Monitoring Service Information 2017.





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
