# Peer review of "What caused the extreme CO concentrations during the 2017 high pollution episode in India?"

_Atmospheric Chemistry and Physics, 2018_

## Referee Comment (RC1) · Anonymous Referee #1 · 21 Dec 2018

This study uses new high-resolution satellite measurements in combination with ground based in situ measurements and regional modeling to diagnose the contributions to extreme CO over northwest India for four days in November 2017. The authors analyze modeled meteorology differences between a polluted period and a cleaner period as well as identify sector-based contributions to the pollution event. The results are timely and interesting, with two very important outcomes – that India now experiences higher pollution than China, and that anthropogenic emissions are important contributors to the high pollution event in India.

The manuscript is well written and fits within the scope of ACP, in particular by analyzing new remote sensing measurements and using regional modeling. The general and broader implications of the study could be discussed to a deeper level, perhaps

with more emphasis on the differences between India and China. Analysis is systematic and thorough, although some areas need clarification. Below are several specific comments, roughly in order of importance. All other comments relate to technical corrections.

**Specific Comments:**

1. Missing emissions
a) Please provide motivation for increasing GFAS emissions by 5 or 10 times. For example, are there reasons to believe these are the bounds on fire emission uncertainty? Add in to methods on page 5 or work into the analysis on page 8.
b) Low emissions are not the only way an inventory can be wrong. With small fires, such as agricultural fires, the issue is that fires could be completely missed (as discussed on page 12, lines 8-12), meaning that increasing emissions will not help (i.e. 10 x 0 = 0). This might be what is occurring for GFAS in this study, especially for 11-14 November. TROPOMI shows many more CO hotspots in NW India between 75 E and 80 E (Fig. 5c), compared to the 5 × GFAS plot (Fig. 5e). In comparison, NW India between 70 E and 75 E seems better captured by the modeled hot-spots. This discussion is currently missing in the manuscript. Potential missing emissions could be further analyzed by spatially comparing the 5 × GFAS to the 1 × GFAS simulation to determine how much the emission increase resulted in CO column increase in the underrepresented area. A difference plot to TROPOMI could help determine the location of missing emissions. Also, a comparison to an inventory like GFED 4s, which includes an algorithm to capture small fires, could be used to analyze the spatial uncertainties in the emissions. As a result, fire emissions required in the underrepresented area could be estimated and assessed for plausibility.
c) Is there reason to believe MACCity is too low by 20%?

2. Selection of the two date periods
a) More motivation is needed for selecting the date periods. The manuscript mentions "based on patterns seen in TROPOMI" (page 7, line 1), and suggests this is presented

in Section 4.3, which is not the case. The ground-based timeseries (Fig. 7) implies the cut-off could be Nov. 11-14, but also Nov. 11-16.

b) Clarify why TROPOMI data before November 11 was not used (was it not available?).

c) Section 3.4 discusses that ground stations show the polluted period begins on November 3rd. Why are the ground-based timeseries only shown from November 11th onwards? It would be valuable to plot all of November in order to see the increase from the 3rd of November onwards. Alternatively, the investigation of ground station data could be informed by TROPOMI and only the same time periods investigated.

3. The paper would benefit from a little more direct discussion on why investigating CO is important - e.g. in addition to a CO health threshold, co-emitted species are also relevant for human health; CO can be measured from space so using this to analyze transport of CO can determine the impact of transported pollution. This would be best built into the Introduction, Discussion and Conclusion. Also, discussing the potential impact on many people in this region would help motivate why a study of this specific region is important. Finally, while meteorology and stagnant conditions seem to drive this pollution event, there would not be as high pollution in the region without the high anthropogenic emissions. The authors could be a little more direct in highlighting this (e.g. page 15, line 12). Quantification of the anthropogenic contribution could be added to the Abstract, Discussion and Conclusion.

4. Expand the discussion of India versus China

a) Using the CAMS reanalysis timeseries of 2012-2017 as November average CO over the IGP and comparing with NE China could add more support to the argument on page 7 that India is now more polluted than Chna. Expanding this discussion would help make the manuscript less focused on a specific area.

b) Page 8, line 1: Please explain that if NCAP does not have emission reduction targets, what does it aim to do?

5. Emission resolution

How are the 0.5 × 0.5 degree MACCity emissions interpolated to the 10 km by 10 km WRF grid? Are they downscaled? Is the result mass conserving?

6. Introduction:

a) Page 1, line 21: What is the temporal resolution of the "high concentrations", i.e. is 10mg/m$^3$ per 8hr, 24 hr? Also describe temporal resolution for 400 mg/m$^3$ on page 2, line 8 so we can compare to the health standard more accurately.

b) In general, the motivation for using both mg/m$^3$ and ppb in the manuscript is unclear. I suggest to make it clearer on page 2, line 8 why the conversion to ppb is described, and to only use ppb from there onwards (e.g. Page 11 swaps back to using mg/m$^3$ even though it talks about plots that are displayed in ppb).

c) Page 2, paragraph starting line 10: Why is GFAS uncertainty brought up in this paragraph, i.e. why is it relevant for this study? I suggest to include an introduction of the fire emission sensitivity tests (× 5, × 10) in the paragraph on page 3.

d) Page 2, lines 27-29: Clarify "this period". Did TROPOMI (and CAMS) observe high mixing ratios for November? All days or certain days?

7. Section 2:

a) More information is needed about the TROPOMI a priori. For example, does it come from a model climatology? Is it spatially and temporally varying?

b) Are land-only retrievals used or ocean as well? If ocean retrievals are used, explain retrievals over ocean (e.g. must have clouds).

c) Page 4, line 14: Biases in the KNMI, 2018 report are relative to MOPITT and IASI which are assimilated into CAMS, so are not independent measurements. Please remove "satellite" from this sentence, leaving the bias relative to TCCON.

d) Define vertical resolution of CAMS rather than "various pressure levels" (page 4, line 16).

e) Define time period of CAMS data used. I interpret that 2012-2017 is used to look at October-December interannual variability, and an earlier version of CAMS was used for November 2017 boundary conditions in WRF.

f) Does WRF nudge completely towards the meteorology, or does it nudge as a percentage?

g) The WRF set-up is unclear. What is the temporal period of WRF modeling? Is there a spin-up period? Are there some WRF with tracers simulations and some WRF-chem simulations? I did not realize until page 13, line 34 that there was a MACCity-only simulation. Is OH oxidation of CO performed in the sensitvity run (a major sink) subtracted from the total CO in all simulations? It would be helpful to define the base-run that everything else is compared with. Perhaps a table recording all the simulations performed would also be helpful.

h) Figure 2: Names which are combinations of region and station are a little confusing. The acronyms are not used further in the manuscript, so I suggest writing the whole name on the map and coloring/shading the regions underneath, or adding boxes around the three regions investigated. Add latitudes and longitudes to the map.

i) Why is the OH climatology scaled by 0.92?

j) Page 7, line 2: Expand how the amount of data per day was used to select periods - e.g. how many data points are in each period, were they chosen to be similar?

8. Section 3:

a) While CAMS has been compared to TROPOMI globally (page 3, line 22) bias and correlation between TROPOMI and CAMS for the IGP region is needed to ensure CAMS can be used to probe the relationship between 2017 and earlier years. A sentence or two on this subject would be a valuable addition.

b) Figure 5: Column and row headings in this figure would make it faster and easier for a reader to compare the plots.

c) I suggest that section 3.3. does not need to be separate to section 3.2, because it is still comparing WRF and TROPOMI.

d) Figure 7 Caption: Plots are in ppb not mg/m$^3$.

9. Section 4:

a) Page 11 lines 1 to 9: This text can be removed because the methodology section

(page 6) mentions that stations close to sources are not used in the analysis.

b) The authors might want to discuss the episodic nature of fire emissions versus the consistent anthropogenic emissions. Also, while a source might contribute a small amount to the total amount of CO, it might be a more substantial contributor for anomalous CO. Finally, in Table 2, why wasn't the period split into 11-14 November and 15-19 November? It is unclear what showing October 1 to November 19 in Table 2 adds to the manuscript, when the focus has been on the two periods.

c) Page 13, line 30: Show the plots of outflow in WRF upper layers to support the argument. A map of the mean meteorological conditions in each period overlaid on the WRF maps of CO may be helpful to visualize the differences between conditions. Alternatively, a Hovmöller diagram of the WRF results may help visualize this outflow.

d) Page 14, lines 7-10: Reword to something such as "In our study, 2017 meteorological conditions are more favorable towards pollution accumulation than 2015 conditions." Also, what data exactly supports this claim? Is it the CAMS concentrations? Or is it meteorological data that isn't shown?

e) Page 14, Section 4.3: Does the inclusion of OH oxidation fix the overestimation for 15-19 Nov in the non-IGP region? I suggest to add another bar to Fig. 6 for the simulation that includes OH oxidation. Section 4.2 would flow better if it was moved to the end of section 3.2.

10. Section 5:

a) Page 15, line 22: Mention the quantitative amount that previous studies have found GFAS to be underestimating fire emissions of CO.

b) Page 15, line 29: Are there any leads on geoengineering meteorology? Also, geoengineering meteorology could have unforeseen consequences for the local or downwind regions, while reducing emissions is likely good overall.

**Technical corrections:**

I suggest to use consistent date formats throughout the manuscript. For example on the first page there are already several formats being used:"13 October 2017" (page

1, line 1); "11 and 19 November 2017" (page 1, line 6); "11-14 November" (page 1, line 11); "15th of November" (page 1, line 11). The changing of formats can be a bit discontinuous for the reader.

Page 1, Line 1: 2017, measures $\longrightarrow$ 2017, has measured

Page 1, line 11: The meteorological situation$\longrightarrow$ Meteorological conditions... were most likely

Page 1, lines 13-14: "...emphasizing the important role of atmospheric dynamics." seems like an unfinished sentence. Expand to explain the role specifically.

Page 2, lines 4-5: Flip the sentence, "Nine out of ten most polluted cities are in the IGP according to the 2018 WHO..." etc.

Page 2, line 4: Define WHO acronym.

Page 2, line 12: Sentence beginning "In addition" is suggested to move to end of paragraph.

Page 2, line 19: practices, these other

Page 2, line 21: extend $\longrightarrow$ extent

Page 2, line 25: measurement instrument $\longrightarrow$ measurements

Page 2, line 26: the orbits of scientific $\longrightarrow$ the scientific

Page 2, line 30: combining $\longrightarrow$ comparing

Page 2, line 32 to page 3, line 2: Unusual to stop the list of four objectives in the middle with a full stop after objective (2). Suggest to re-write.

Page 3, line 6: section also the role of met... $\longrightarrow$ section the role of meteorological conditions is also...

Page 3, line 15: Define SICOR acronym.

Page 4, line 6: global resolution $\longrightarrow$ global horizontal resolution

Page 4, line 20: Remove "ARW" if not used again.

Page 4, line 24: Remove "MYJ" if not used again.

Page 4, line 24: Should "Eta" be capitalized?

Page 4, line 27: Add citation after RRTM and add ", respectively" to the end of the sentence.

Page 6, line 17: As outlined in section 2.1, the...

Page 7, line 6: ...regions (Fig. 2 red and black labels are inner and outer city, respectively).

Page 7, lines 12-13: ...China was the most polluted...

Page 8, lines 3-4: To determine how unique these high CO values were, we analysed the last four years of CAMS data.

Page 8, line 8: However, ⟶ These

Page 8, Line 25: Reword the last sentence to say something such as "Atmospheric chemistry is not sufficient to explain differences in CO between the two time periods."

Page 9, line 4: then ⟶ than

Page 11, Figure 8 caption: (b.) total column mixing ratio.

Page 13, line 21: According to Fig. 8, the wind speeds clearly...

---

## Referee Comment (RC2) · Anonymous Referee #2 · 17 Jan 2019

Review of Dekker et al., "What caused the extreme CO concentrations during the 2017 high pollution episode in India?"

Dekker et al. use a combination of satellite retrievals, chemical-transport model simulations, and surface observations to explore why North India experienced extreme CO concentrations during a recent pollution episode that received much popular attention. During October and November, North India frequently experiences spectacularly high concentrations of CO, PM2.5, and other primary pollutants. Public discussion generally focuses on crop-stubble (and other biomass) burning as the principal culprit. Here, Dekker et al provide evidence that this assumption may be simplistic and incorrect: instead, they suggest that poorly diluted primary CO emissions from other sources (transport, industry) etc. explains the episode.

[Figure]

The paper is generally clearly written and the caliber of the technical work is appropriate. I would recommend the manuscript for publication in ACP with minor edits.

The authors suggest that meteorology is a dominant explanation for the CO episodes. What I found lacking in the argument provided here was in explaining the genesis/formation of the episode. Was there a sudden shift in meteorology (e.g., reduction in wind speeds and PBLH) that coincided with the initiation of the pollution episode? Here, I concur with what Reviewer #1 has suggested, namely regarding a more detailed explanation of why the specific time periods included here were chosen for analysis. If it is possible, showing a longer time period that illustrates the formation of the episode could be helpful.

To strengthen your argument, you might also appeal more directly to the existing emissions inventories. For example, it would be useful to quantify in a table or figure the share of primary emissions (for this season) across the IGP that are attributable to the various sectors and to crop burning. You might consider distinguishing here between fire- and non-fire periods, perhaps with a sensitivity case that allows for the possibility you alluded to that fire CO emissions are underestimated.

Other Minor suggestions:

* Units: the manuscript switches frequently between using mass concentration ($\mu$g/m3, mg/m3) units and mixing ratio units (ppb) for CO. While I recognize that there are good reasons for using these alternative unit systems, it would be helpful to orient readers by providing an approximate conversion factor and when possible crafting the core narrative of the paper around one unit system).

* Health-based standards: CO is subject to national ambient air quality standards in India. It would be helpful to compare the observed concentrations to Indian guidelines and other relevant values (ie., Chinese, US, EU). I believe that Indian standards may be more stringent than most other countries for CO – unlike Indian standards for PM2.5, which tend to be quite permissive.

* Section 4.2: the role of adverse meteorology in driving North Indian pollution episodes is widely appreciated in the literature. I would suggest looking at (and citing when appropriate) the following articles:

* Guttikunda and Gurjar, "Role of meteorology in seasonality of air pollution in megacity Delhi, India": doi: 10.1007/s10661-011-2182-8

* Gani et al., "Submicron aerosol composition in the world's most polluted megacity: The Delhi Aerosol Supersite campaign", doi: 10.5194/acp-2018-1066

* Tiwari et al., "Aerosol optical properties and their relationship with meteorological parameters during wintertime in Delhi, India", doi:10.1016/j.atmosres.2014.10.003

* Tiwari et al., "Variability in atmospheric particulates and meteorological effects on their mass concentrations over Delhi, India", doi: 10.1016/j.atmosres.2014.03.027

* From the various comparisons made with conditions in China, it is evident that the authors are somewhat surprised by the severity of the pollution problem in North India. Yet the magnitude of the PM2.5 problem in North India is becoming quite well documented, especially in literature focused on remotely sensed PM2.5 levels. The evidence is now quite clear that India has overtaken China for population-weighted PM2.5 levels, while other large countries (Bangladesh, Pakistan, Nigeria, Egypt) can be even more polluted. In my view, the focus on China's pollution in the popular imagination – even when nowhere near the highest in the world – arises in part because of the rapid improvement in the availability of ground-based data there. See for example Shaddick et al, ES&T 2018: https://pubs.acs.org/doi/pdf/10.1021/acs.est.8b02864

* A discussion of data quality and measurement uncertainty for the in-situ CPCB pollution measurements would be appropriate. Do the surface measurements appear to be generally well calibrated and reliable?

---

## Author Comment (AC1) · 20 Feb 2019

**Below, first our response to Anonymous Referee #1, followed by our response to Anonymous Referee #2.**

**Response to the review of Anonymous Referee #1**

*Dear referee, thank you very much for your very thorough and helpful review of our paper!*
*We will discuss below our changes in the paper and/or our thoughts about the points that you mention in your review in blue. We wrote our comments in italic and the improved text within quotes.*

1. Missing emissions
a) Please provide motivation for increasing GFAS emissions by 5 or 10 times. For example, are there reasons to believe these are the bounds on fire emission uncertainty?
Add in to methods on page 5 or work into the analysis on page 8.

*The exact uncertainty in the fire emissions from the database is not known. The papers of Mota and Wooster, (2018) Cusworth et al. (2018) and Huijnen et al. (2016) indicated all that there very likely is a (large) underestimation of the fire emission databases that are currently existent. {Mota:2018uj} mentions a 64% underestimation of particulate matter (PM) emitted in GFEDv4.1. Cusworth mentions that compared to the GFED+agriculture emissions, GFEDv4.1 PM emissions are a factor 3 too low.*
*For CO we could not find an estimate of the underestimation, but through personal contact with Dr. Pankaj Sadavarte (now SRON, previously at Indian Institute of Technology Bombay, IITB) we learned that they use a regional emission inventory at IITB in which the emissions of CO were a factor 2.7 times the GFAS estimate of CO emissions in November and the emissions might even be higher.*

*To make this more clear, we added in the text, section 2.3:* "However, there are strong signs that fire emission inventory datasets, such as GFAS and GFED, do not capture all of the biomass burning emissions (Mota and Wooster, 2018; Cusworth et al., 2018; Huijnen et al., 2016). Also, the conversion of fire occurrence to CO emissions depends on factors such as combustion efficiency, biome type, and soil characteristics, which adds uncertainty to the emission estimates (Werf van der et al., 2010). For PM, it is estimated that the GFAS emissions are underestimated approximately a factor 2-6 (Mota and Wooster, 2018; Cusworth et al., 2018; Huijnen et al., 2016). For CO, less information is available, but similar underestimates are expected."

b) Low emissions are not the only way an inventory can be wrong. With small fires, such as agricultural fires, the issue is that fires could be completely missed (as discussed on page 12, lines 8-12), meaning that increasing emissions will not help (i.e. 10 x 0 = 0). This might be what is occurring for GFAS in this study, especially for 11-14 November. TROPOMI shows many more CO hotspots in NW India between 75 E and 80 E (Fig. 5c), compared to the 5 _ GFAS plot (Fig. 5e). In comparison, NW India between 70 E and 75 E seems better captured by the modeled hot-spots. This discussion is currently missing in the manuscript. Potential missing emissions could be further analyzed by spatially comparing the 5 _ GFAS to the 1 _ GFAS simulation to determine how much the emission increase resulted in CO column increase in the underrepresented area. A difference plot to TROPOMI could help determine the location of missing emissions. Also, a comparison to an inventory like GFED 4s, which includes an algorithm to capture small fires, could be used to analyze the spatial uncertainties in the emissions. As a result, fire emissions required in the underrepresented area could be estimated and assessed for plausibility.
*Yes, it is true that some fires might be missed. We have been looking into this and have done simulations based on fire radiative power in the Suomi-NPP Visible Infrared Imaging Radiometer Suite (VIIRS) which has a higher resolution and better sensitivity to fires. Furthermore, simulations have been performed based on a land use map, as an alternative method to locate crop stubble burning in November. Unfortunately, the GFED 4s data is not yet available for this recent time*

*period. The VIIRS and land-used based simulations, indicated that although it is quite likely that some small burning areas are missed in the GFAS data, the resolution of the satellite and model are relatively low for this purpose and the spreading over the IGP area went relatively quick, making the exact location of the emissions not very important. So, on this resolution, increasing the emissions at the GFAS locations is probably sufficient to account for missing emission locations.*
*We added in the conclusions: "Fires missed by GFAS observations might explain part of this increase." We added in section 4.1: "Or a substantial amount of fires is missed in GFAS in this period."*

c) Is there reason to believe MACCity is too low by 20%?
*No, but there is quite a large uncertainty in the data, estimated at 50%-200% as mentioned in the paper. So, 20% too low is still well within the uncertainties.*

2. Selection of the two date periods
a) More motivation is needed for selecting the date periods. The manuscript mentions
"based on patterns seen in TROPOMI" (page 7, line 1), and suggests this is presented in Section 4.3, which is not the case. The ground-based timeseries (Fig. 7) implies the
cut-off could be Nov. 11-14, but also Nov. 11-16.
*Section 4.3 is only mentioned to refer to the weather conditions. We agree that based on the weather conditions only, the split could have been on November 15 or 16 also. By choosing two periods of four days, both periods have sufficient data to compare. We clarified this as follows: "We also averaged over several days of data, concentrating on two periods: 11-14 November 2017, and 15-19 November 2017, in order to obtain a gap free image of Northern India (18 November no TROPOMI data was available). We selected these two periods based on the patterns seen in TROPOMI data, the equal number of days per period, and the weather conditions (see 4.3)."*

b) Clarify why TROPOMI data before November 11 was not used (was it not available?).
*On 9 November the first radiance measurements were done. The near nominal temperature was first reached on November 11 and so considered as the first day with reliable data. We added in the text:* "The instrument reached near-nominal temperatures on 11 November, which is considered the first day of reliable data from TROPOMI (Borsdorff et al., 2018a)"

c) Section 3.4 discusses that ground stations show the polluted period begins on November 3rd. Why are the ground-based timeseries only shown from November 11[th] onwards? It would be valuable to plot all of November in order to see the increase from the 3rd of November onwards. Alternatively, the investigation of ground station data could be informed by TROPOMI and only the same time periods investigated.
*We chose to show the ground-based time series from 11 November onwards to make the data periods more consistent, because for these days also TROPOMI data is available. It is however important to consider that we saw in the longer timeseries a build-up of CO levels from October on. This can be seen in the data series of 2017 in Fig. 4 as well. We changed the first sentence of this section to clarify this:* "CO concentrations at the ground-level measurement stations that are used are generally increasing until 14 November, compared to earlier measurements. This is in accordance with the total column CO levels seen in the 2017 timeseries from CAMS."

3. The paper would benefit from a little more direct discussion on why investigating CO is important - e.g. in addition to a CO health threshold, co-emitted species are also relevant for human health; CO can be measured from space so using this to analyze transport of CO can determine the impact of transported pollution. This would be best built into the Introduction, Discussion and Conclusion. Also, discussing the potential impact on many people in this region would help motivate why a study of this specific region is important. Finally, while meteorology and stagnant conditions seem to drive this pollution event, there would not be as high pollution in the region without the high anthropogenic emissions. The authors could be a little more direct in highlighting this (e.g. page 15, line 12). Quantification of the anthropogenic contribution could be added

to the Abstract, Discussion and Conclusion.

*We added to the Introduction: "The total population in the IGP region (including parts of Pakistan, Bangladesh and Nepal) exceeds 400 million and is growing. This means that the bad air quality is effecting hundreds of millions of people for a large part of the year. This makes it very important to investigate the origin and transport of pollution in the area. Due to its lifetime of several weeks, CO can also be used as a proxy of other, co-emitted, anthropogenic pollution."*

*We added to the Discussion: "This means that not only the people living in Delhi, but also a large part of the hundreds of millions of people inhabiting the rest of the IGP are affected by the bad air quality."*

*We added to the Conclusion: "This demonstrates the high importance of investigating the sources and the transport of pollution as hundreds of millions of people are living in the IGP and their health is likely affected by the bad air quality. CO is very suitable for investigating air pollution, not only because of the negative health impact of CO itself, but also as tracer to track the dispersion of other pollutants, due to its life-time of several weeks."*

4. Expand the discussion of India versus China
a) Using the CAMS reanalysis timeseries of 2012-2017 as November average CO over the IGP and comparing with NE China could add more support to the argument on page 7 that India is now more polluted than Chna. Expanding this discussion would help make the manuscript less focused on a specific area.

*Thanks for the suggestion. We have also thought about expanding the discussion to include China, but decided to keep the focus on India. We think that when extending this discussion more to China, we also should include more discussion on the sources of CO emission in China and the meteorological situation in China. This would distract too much from the message we want to convey with this paper: the high pollution levels in India and the sources thereof (and thus the possible solutions to decrease these high pollution levels). In China, already quite some effort has been and will be spend to decrease pollution levels.*

b) Page 8, line 1: Please explain that if NCAP does not have emission reduction targets, what does it aim to do?

*NCAP is more focused on making an inventory of how the air quality is at the moment and making plans on how to best reduce the emission. We added the following to the text (last part of the sentence):*

"NCAP does not yet include strict targets for emission reductions and rather focusses on setting up an effective ambient air quality monitoring network and making plans for prevention, control and abatement of air pollution."

5. Emission resolution
How are the 0.5 _ 0.5 degree MACCity emissions interpolated to the 10 km by 10 km WRF grid? Are they downscaled? Is the result mass conserving?

*The emissions are bilinearly interpolated. The result is not fully mass conserving, but the resulting differences are very small and much smaller than the error of the emission uncertainty.*

6. Introduction:
a) Page 1, line 21: What is the temporal resolution of the "high concentrations", i.e. is $10mg/m^3$ per 8hr, 24 hr? Also describe temporal resolution for 400 mg/m3 on page 2, line 8 so we can compare to the health standard more accurately.

We added the time resolution: "carbon monoxide with values of up to 10 mg/m$^3$ several days in a row". And: *"reaching 15 minute average values up to 400 mg/m$^3$".*

b) In general, the motivation for using both mg/m3 and ppb in the manuscript is unclear. I suggest to make it clearer on page 2, line 8 why the conversion to ppb is described, and to only use ppb from there onwards (e.g. Page 11 swaps back to using mg/m3 even though it talks about plots that are displayed in ppb).

*Thanks for noticing. The mg/m$^3$ was intended to make it easier to compare to the standards. But we decided indeed to look further at ppb levels, to make the CO levels less dependent on orography and to make total columns easier to compare with ground-level CO levels. The mg/m$^3$ on page 11 was meant to be ppb already. We will change this in the text and added: "... we will use ppb from here on".*

c) Page 2, paragraph starting line 10: Why is GFAS uncertainty brought up in this paragraph, i.e. why is it relevant for this study? I suggest to include an introduction of the fire emission sensitivity tests (_ 5, _ 10) in the paragraph on page 3.
*We moved the following sentence from the introduction to the paragraph explaining the WRF inputs and the 1,5,10 sensitivity test: "However, there are strong signs that these datasets do not capture all of the biomass burning emissions (Mota and Wooster, 2018; Cusworth et al., 2018; Huijnen et al., 2016). Also, the conversion of fire occurrence to CO emissions depends on factors such as combustion efficiency, biome type, and soil characteristics, which adds uncertainty to the emission estimates (Werf van der et al., 2010).".*

d) Page 2, lines 27-29: Clarify "this period". Did TROPOMI (and CAMS) observe high mixing ratios for November? All days or certain days?
*We changed the previous sentence to: "TROPOMI observed very high column mixing ratios over the Northern part of India from November 11 - 19".*

7. Section 2:
a) More information is needed about the TROPOMI a priori. For example, does it come from a model climatology? Is it spatially and temporally varying?
*When the right regularization is used, as is done for the TROPOMI data, the TROPOMI CO a priori is not relevant in the retrieved data. We explain this now more extensive:*
"... According to Borsdorff et al. (2014):
$$C_{retr} = C_{prior} + AK(C_{true} - C_{prior}) + e_x \qquad (1)$$
where ex represents the error on the retrieved trace gas profile. The Equation simplifies to Eq. 2:
$$C_{retr} = AK(C_{true}) + e_x \qquad (2)$$
when the effective null space contribution of the a priori profile is eliminated, which is the true for the chosen regularisation parameter for the TROPOMI CO data, as is explained in Borsdorff et al. (2014)

b) Are land-only retrievals used or ocean as well? If ocean retrievals are used, explain retrievals over ocean (e.g. must have clouds).
*Both land and ocean retrievals are used, ocean retrievals indeed require clouds. We explain already all the filtering we apply on the data in the TROPOMI subsection of Data and methods. We added the following sentence to make it more clear that also some data over sea was included: " This retrieval method allowed to include some measurements over sea with low-level clouds (Borsdorff et al., 2018c)."*

c) Page 4, line 14: Biases in the KNMI, 2018 report are relative to MOPITT and IASI which are assimilated into CAMS, so are not independent measurements. Please remove "satellite" from this sentence, leaving the bias relative to TCCON.
*Good point. We removed the word "satellite".*

d) Define vertical resolution of CAMS rather than "various pressure levels" (page 4, line 16).
*The pressure levels are the following: "1000    950    925    900    850    800    700    600    500    400    300    250    200    150    100    70    50    30    20    10    7    5    3    2    1"*
*Describing them all would not add much useful information, compared to the amount of text needed. We changed "various pressure levels" to "25 pressure levels". In the link following the text, the pressure levels can be found.*

e) Define time period of CAMS data used. I interpret that 2012-2017 is used to look at October-December interannual variability, and an earlier version of CAMS was used for November 2017 boundary conditions in WRF.
*We used 2012-2017 CAMS total column data for the interannual variability and the CAMS data for the pressure levels for October and November 2017 for the WRF boundary conditions.*
*We added October and November in the description of the WRF data: "The boundary conditions for CO came from the CAMS CO October and November data on pressure levels, interpolated to the WRF model levels." We added the following sentence to the CAMS subsection: "September-January total column CAMS data for the years 2012-2017 is used in section 3.1."*

f) Does WRF nudge completely towards the meteorology, or does it nudge as a percentage?
*Only partly: WRF nudges the interior meteorology towards the values prescribed at the lateral boundary conditions in a "relaxation zone" of, in our case, 5 cells outside the WRF grid to prevent very sharp changes in the meteorology.*

g) The WRF set-up is unclear. What is the temporal period of WRF modeling? Is there a spin-up period? Are there some WRF with tracers simulations and some WRF-chem simulations? I did not realize until page 13, line 34 that there was a MACCity-only simulation. Is OH oxidation of CO performed in the sensitvity run (a major sink) subtracted from the total CO in all simulations? It would be helpful to define the base-run that everything else is compared with. Perhaps a table recording all the simulations performed would also be helpful.
*We run our model in all cases for two months: October and November 2017. October 1 to November 11 can thus be considered as spin-up period. We did not do a full WRF-Chem simulation, we only use its tracer function. OH oxidization is only taken into account in the sensitivity run, not in the other simulations. Due to the outflow out of the domain, part of the CO is already decreasing. We made this more clear in the text now: "Different CO emission inventories are available for Southern Asia. As in CAMS, we used MACCity anthropogenic CO emissions for the year 2017 at a resolution of 0.5°x0.5°(Lamarque et al., 2010). We implemented nine different CO tracers representing the MACCity emission categories in WRF-Chem (see Table 1). The MACCity database estimates worldwide monthly emission strengths for these emission categories. An additional tracer was used to account for CO transported from the CAMS derived boundary conditions: we refer to this CO tracer as background in this paper (Table 1). For biomass burning emissions, we used GFAS data with a resolution of 0.1°x0.1°(available for download from: http://apps.ecmwf.int/datasets/data/cams-gfas/). However, there are strong signs that fire emission inventory datasets, such as GFAS and GFED, do not capture all of the biomass burning emissions (Mota and Wooster, 2018; Cusworth et al., 2018; Huijnen et al., 2016). Also, the conversion of fire occurrence to CO emissions depends on factors such as combustion efficiency, biome type, and soil characteristics, which adds uncertainty to the emission estimates (Werf van der et al., 2010). For PM, it is estimated that the GFAS emissions are underestimated approximately a factor 2-6 (Mota and Wooster, 2018; Cusworth et al., 2018; Huijnen et al., 2016). For CO, less information is available, but similar underestimates are expected. In our base setup, we ran the model for the period 1 October - 20 November, including the tracers listed in Table 1, including the original GFAS emissions and the background. In this paper, results are shown for November 11-20, October is considered as a spin-up period. Besides the base run, we did some extra simulations including 0 (referred to as: MACCity), 1 (base setup), 5 (+5GFAS) and 10 (+10GFAS) times the original GFAS emissions, based on the estimated underestimation of fire PM emissions."*

h) Figure 2: Names which are combinations of region and station are a little confusing. The acronyms are not used further in the manuscript, so I suggest writing the whole name on the map and coloring/shading the regions underneath, or adding boxes around the three regions investigated. Add latitudes and longitudes to the map.

*We now included latitudes and longitudes to the map and wrote the full station locations name instead of acronyms. There are no further regions than the station locations indicated with the dots.*

[Figure]

i) Why is the OH climatology scaled by 0.92?
*The OH field has been optimized using methylchloroform data, leading to a scaling factor of 0.92 (Krol et al., 2013).*
*We changed the text to:* "and the corresponding OH climatology, based on Spivakovsky et al. (2000), scaled by 0.92 (Huijnen et al., 2016, 2010; Krol et al., 2013)"

j) Page 7, line 2: Expand how the amount of data per day was used to select periods -
e.g. how many data points are in each period, were they chosen to be similar?
*Yes, both periods contain 4 days of data this way. We changed the text to:* "We also averaged over several days of data, concentrating on two periods: 11-14 November 2017, and 15-19 November 2017, in order to obtain a gap free image of Northern India (18 November no TROPOMI data was available). We selected these two periods spanning an equal number of days, based on the patterns seen in TROPOMI data and the changing weather conditions (see 4.3)"

8. Section 3:
a) While CAMS has been compared to TROPOMI globally (page 3, line 22) bias and
correlation between TROPOMI and CAMS for the IGP region is needed to ensure
CAMS can be used to probe the relationship between 2017 and earlier years. A sentence
or two on this subject would be a valuable addition.
*We added the following sentence:* "For the India region, a 2.9% difference was found with CAMS with a standard deviation of 6% and a Pearson correlation coefficient of 0.9 (Borsdorff et al., 2018a)"

b) Figure 5: Column and row headings in this figure would make it faster and easier for
a reader to compare the plots.
*We added the headers now per subfigure.*

[Figure]

c) I suggest that section 3.3. does not need to be separate to section 3.2, because it is
still comparing WRF and TROPOMI.
*We prefer to keep these sections separated to highlight the findings presented here. As a separate
section, it appears clearer to the reader that this is an important message of our paper. We now
divided 3.2 in two subsections, 3.2.1: "Agreement between WRF and TROPOMI", and 3.2.2 is what
was section 3.3 before.*

d) Figure 7 Caption: Plots are in ppb not mg/m₃.
*Changed the caption.*

9. Section 4:
a) Page 11 lines 1 to 9: This text can be removed because the methodology section (page 6) mentions
that stations close to sources are not used in the analysis.

*We wanted to make clear that in the excluded stations, some high CO mixing ratios were found and that we are not sure what the cause thereof is. We now removed the lines 1-9 in this section and added a more general statement to the next subsection on source attribution: "Some ground stations were excluded because of the proximity to large CO sources, leading to very high observed CO concentrations. Although these measured enhancements are likely traffic related, it is possible that some of the enhancements are caused by local biomass burning. Overall, however, our results point to a relatively small role of biomass burning in the enhanced CO concentrations over the largest part of the IGP, as shown in Fig. 8 for inner-city stations. A similar contribution of biomass burning is found out of the city (not shown)."*

b) The authors might want to discuss the episodic nature of fire emissions versus the consistent anthropogenic emissions. Also, while a source might contribute a small amount to the total amount of CO, it might be a more substantial contributor for anomalous CO. Finally, in Table 2, why wasn't the period split into 11-14 November and 15-19 November? It is unclear what showing October 1 to November 19 in Table 2 adds to the manuscript, when the focus has been on the two periods.

*We agree that the episodic nature of fire CO emissions can be underlined more in the paper. In Table 2, we deliberately show not only the average but also maximum values to represent the CO variability.*
*We added the following sentence to the start of the section: "The CO emission contribution from fires has certainly a more episodic nature than the anthropogenic sources in MACCity. In this section, we do not only include the average contribution, but also consider the maximum contribution of fire emissions to investigate short-term variations in the contribution of fires to CO."*
*In our opinion splitting this period into 11-14 and 15-19 November does not add extra information in the discussion of the contribution of the fire emissions, which is similar in both periods.*
*We included the longer period, because the fire emissions before November 11 have been higher, and we wanted to give a fair comparison including also the higher contribution. We included now the following sentence: "The steadily increase in CO levels started before November 11; to make sure we did not miss part of the biomass burning contribution, we included in Table 2 also a longer period of the post-monsoon season: 1 October - 19 November."*

c) Page 13, line 30: Show the plots of outflow in WRF upper layers to support the argument. A map of the mean meteorological conditions in each period overlaid on the WRF maps of CO may be helpful to visualize the differences between conditions. Alternatively, a Hovmöller diagram of the WRF results may help visualize this outflow.

d) Page 14, lines 7-10: Reword to something such as "In our study, 2017 meteorological conditions are more favorable towards pollution accumulation than 2015 conditions." Also, what data exactly supports this claim? Is it the CAMS concentrations? Or is it meteorological data that isn't shown?
*The main point is not that 2017 is significantly different from 2015, which we can't prove by comparing the available data, but that both 2015 and 2017 show the build-up of air pollution. From the data that we have, 2017 seems slightly worse than 2015. We changed the text as follows: "From the meteorological and observational data we conclude that 2017 shows a similar build of pollution as in the 2015 study, however, circumstances for CO accumulations were slightly stronger in November 2017. In our study, as in Verma et al. (2017), stagnant weather conditions are an important cause of the high pollution levels."*

e) Page 14, Section 4.3: Does the inclusion of OH oxidation fix the overestimation for 15-19 Nov in the non-IGP region? I suggest to add another bar to Fig. 6 for the simulation that includes OH oxidation. Section 4.2 would flow better if it was moved to the end of section 3.2.

*No, the inclusion of OH oxidation does not fix the overestimation. We thought about adding an extra bar to Figure 6, but we think it will complicate the Figure and discussion more than necessary. We also would like to keep the section in the discussion and not in the results.*

10. Section 5:
a) Page 15, line 22: Mention the quantitative amount that previous studies have found GFAS to be underestimating fire emissions of CO.
*We added these values at the end of the sentence now:* "The difference could be accounted for by increasing the GFAS emissions by 500%-1000%, a rather large increase compared to the 200%-600% increase found in the last named studies"

b) Page 15, line 29: Are there any leads on geoengineering meteorology? Also, geoengineering meteorology could have unforeseen consequences for the local or downwind regions, while reducing emissions is likely good overall.
*Yes, we agree with this and would definitely not support geo-engineering. As we are no expert on this field, we think it is best to leave the geo-engineering part out in total. We changed the sentence now to:* "Meteorology is found to be key driver of the extreme pollution episodes, however, in conjunction with strong CO emissions."

Technical corrections:
I suggest to use consistent date formats throughout the manuscript. For example on the first page there are already several formats being used:"13 October 2017" (page 1, line 1); "11 and 19 November 2017" (page 1, line 6); "11-14 November" (page 1, line 11); "15th of November" (page 1, line 11). The changing of formats can be a bit discontinuous for the reader.
*We changed the dates now to be consistent as "11-14 November"*

Page 1, Line 1: 2017, measures -> 2017, has measured
*Done*
Page 1, line 11: The meteorological situation -> Meteorological conditions... were most likely
*Done*
Page 1, lines 13-14: "...emphasizing the important role of atmospheric dynamics."
seems like an unfinished sentence. Expand to explain the role specifically.
*We split the sentence and added some extra explanation:* "This emphasizes the important role of atmospheric dynamics in determining the air quality conditions at ground-level and in the total column."
Page 2, lines 4-5: Flip the sentence, "Nine out of ten most polluted cities are in the IGP according to the 2018 WHO..." etc.
*Done*
Page 2, line 4: Define WHO acronym.
*Done*
Page 2, line 12: Sentence beginning "In addition" is suggested to move to end of paragraph.
*Done*
Page 2, line 19: practices, these other
*Done*
Page 2, line 21: extend -> extent
*Done*
Page 2, line 25: measurement instrument -> measurements
*Done*
Page 2, line 26: the orbits of scientific -> the scientific
*Done*
Page 2, line 30: combining -> comparing
*Done*

Page 2, line 32 to page 3, line 2: Unusual to stop the list of four objectives in the middle with a full stop after objective (2). Suggest to re-write.

*We rewrote the sentences so all objectives are now in one sentence. While doing this we identified an extra objective: "We assess this according to our five objectives: (1) whether TROPOMI is in accordance with ground-based measurements and (2) how well WRF is able to reproduce these data; how the pollution is dispersing over India (3), the role of meteorology in the accumulation and transport of CO (4), and shedding more light on sources contributing most to the high pollution over the Indo-Gangetic Plain (IGP) of India (5) in support of future pollution mitigation efforts."*

Page 3, line 6: section also the role of met... -> section the role of meteorological conditions is also...

*Done*

Page 3, line 15: Define SICOR acronym.

*Done*

Page 4, line 6: global resolution -> global horizontal resolution

*Done*

Page 4, line 20: Remove "ARW" if not used again.

*Done*

Page 4, line 24: Remove "MYJ" if not used again.

*Done*

Page 4, line 24: Should "Eta" be capitalized?

*No, it usually is written with all small letters in this context*

Page 4, line 27: Add citation after RRTM and add ", respectively" to the end of the sentence.

*We removed RRTM and placed the citation at that place. We added respectively to the end of the sentence.*

Page 6, line 17: As outlined in section 2.1, the...

*Done*

Page 7, line 6: ...regions (Fig. 2 red and black labels are inner and outer city, respectively).

*"regions (Fig. 2 red and black labels indicate inner and outer city, respectively)*

Page 7, lines 12-13: ...China was the most polluted...

*Done*

Page 8, lines 3-4: To determine how unique these high CO values were, we analysed the last four years of CAMS data.

*Changed to: "To determine how unique these high CO values were during this time of the year over the IGP, we analysed the last four years of CAMS data."*

Page 8, line 8: However, ->These

*Done*

Page 8, Line 25: Reword the last sentence to say something such as "Atmospheric chemistry is not sufficient to explain differences in CO between the two time periods."

*We would like to keep it clear that we did not include the atmospheric chemistry in our WRF base setup. We changed the sentence to: "The simulations in WRF were not including the atmospheric chemistry, but this is probably only playing a minor role (see section 4.3)."*

Page 9, line 4: then -> than

*Done*

Page 11, Figure 8 caption: (b.) total column mixing ratio.

*Done*

Page 13, line 21: According to Fig. 8, the wind speeds clearly...

*Changed to: "As can be seen in Fig. 8, the wind speeds clearly…"*

**Response to the review of Anonymous Referee #2**

*Thank you, referee #2 for your review! We improved our paper with your helpful suggestions, as indicated below in blue. Our comments are written in italic, and the improved text is within quotes.*

The authors suggest that meteorology is a dominant explanation for the CO episodes. What I found lacking in the argument provided here was in explaining the genesis/formation of the episode. Was there a sudden shift in meteorology (e.g., reduction in wind speeds and PBLH) that coincided with the initiation of the pollution episode? Here, I concur with what Reviewer #1 has suggested, namely regarding a more detailed explanation of why the specific time periods included here were chosen for analysis. If it is possible, showing a longer time period that illustrates the formation of the episode could be helpful.

*As can be seen in Fig. 4, total column CO levels over the IGP are rising from the first of October till the 13$^{th}$ of November. For TROPOMI, there are only data available from 11-17 and at 19 November. To have equal amounts of data in two periods of higher and lower CO mixing ratios we chose to have the first 4 days of TROPOMI data in the period of high CO levels and 4 days of lower CO levels 15,16,17, and 19 November. The meteorology also clearly changes around the 15$^{th}$ of November. As can be seen in Figure 9, the wind speed is largely increased from 16 November on, the surface pressure starts to increase from 13 November on, the boundary layer height is high on November 14 and from 18 November on. We described the choice of our periods clearer in the text now:* "We averaged over several days of data, concentrating on two periods: 11-14 November 2017, and 15-19 November 2017, in order to obtain a gap free image of Northern India (18 November no TROPOMI data was available). We selected these two periods spanning equal number of days based on the patterns seen in TROPOMI data and the weather conditions (see 4.3)."

To strengthen your argument, you might also appeal more directly to the existing emissions inventories. For example, it would be useful to quantify in a table or figure the share of primary emissions (for this season) across the IGP that are attributable to the various sectors and to crop burning.

*We already included Figure 8 that shows the share of the primary emissions over the IGP and in Table 2 we focused on the share of fire emissions therein. In the text:* "According to the tracer simulations in our model that are based on MACCity and GFAS emissions, more than 50% of the CO near ground-level in November was caused by residential and commercial combustion. Other main sources are industrial combustion and traffic."

You might consider distinguishing here between fire- and non-fire periods, perhaps with a sensitivity case that allows for the possibility you alluded to that fire CO emissions are underestimated.

*The problem with distinguishing between "fire- and non-fire" periods is that we have a limited amount of data available (TROPOMI: 11-19 November) and in the fire data that we have, there are probably lacking fire CO sources (risking to mark fire days as non-fire days).*

Other Minor suggestions:
* Units: the manuscript switches frequently between using mass concentration (μg/m3, mg/m3) units and mixing ratio units (ppb) for CO. While I recognize that there are good reasons for using these alternative unit systems, it would be helpful to orient readers by providing an approximate conversion factor and when possible crafting the core narrative of the paper around one unit system).

*We stick with the ppb now, except from the introduction on guidelines for CO on ground-level, where we include mg/m3 with a conversion factor.*

\* Health-based standards: CO is subject to national ambient air quality standards in India. It would be helpful to compare the observed concentrations to Indian guidelines and other relevant values (ie., Chinese, US, EU). I believe that Indian standards may be more stringent than most other countries for CO – unlike Indian standards for PM2.5, which tend to be quite permissive.

*This is quite interesting. We weren't aware that this is the case. We looked up the standards to find out that the Indian standards indeed are much more stringent:*

*The Indian standard is 2 mg/m$^3$ for 8 hours,  4 mg/m$^3$ for 1 hour,*

*China:*     *10 mg/m$^3$/hour,  4 mg/m$^3$ for 1 hour.*

*EU:*      *10 mg/m$^3$ for 8 hours.*

*US:*      *10 mg/m$^3$ for 8 hours, 40 mg/m$^3$ for 1 hour.*

*WHO:*     *10 mg/m$^3$ for 8 hours, 30 mg/m$^3$ for 1 hour.*

*Based on:*

*https://acm.eionet.europa.eu/reports/docs/ETCACM_TP_2016_10_AAQstandards.pdf*

*On the other hand, the standard is of course sometimes just a number, if there is no consequence of exceeding the standards (as far as we know, there is not yet any kind of penalty associated with violating these standards in India), a strict standard is not any better than a loose standard.*

*We therefor don't think that adding the discussion of the standards in this paper will add something useful, but we added the Indian standards in the introduction: "At several ground-based measurement stations in the IGP maintained by the Central Pollution Control Board (CPCB, http://cpcb.nic.in/), carbon monoxide (CO) levels amply exceeded the world health organization guidelines (100 mg/m$^3$ for 15 minutes, 10 mg/m$^3$ for 8 hour) and the more stringent Indian standards for CO (2 mg/m$^3$ for 8 hours, 4 mg/m$^3$ for 1 hour) during several days in November 2017 reaching 15 minute average values up to 400 mg/m$^3$"*

 \* Section 4.2: the role of adverse meteorology in driving North Indian pollution episodes is widely appreciated in the literature. I would suggest looking at (and citing when appropriate) the following articles:
\* Guttikunda and Gurjar, "Role of meteorology in seasonality of air pollution in megacity Delhi, India": doi: 10.1007/s10661-011-2182-8
\* Gani et al., "Submicron aerosol composition in the world's most polluted megacity: The Delhi Aerosol Supersite campaign", doi: 10.5194/acp-2018-1066
\* Tiwari et al., "Aerosol optical properties and their relationship with meteorological parameters during wintertime in Delhi, India", doi:10.1016/j.atmosres.2014.10.003
\* Tiwari et al., "Variability in atmospheric particulates and meteorological effects on their mass concentrations over Delhi, India", doi: 10.1016/j.atmosres.2014.03.027

*Thank you for referring to these papers. We added a bit of extra discussion based on these papers and referred to them now:*

*Introduction: "In addition, post-monsoon meteorological conditions can lead to an accumulation of pollutants in Northern India (Liu et al., 2018; Gani et al., 2018; Tiwari et al., 2015, 2014; Guttikunda and Gurjar, 2011)."*

*Meteorological conditions: "Several studies have been performed studying this relation for Delhi based on PM (e.g. Gani et al., 2018; Tiwari et al., 2015;Guttikunda and Gurjar, 2011). Guttikunda and Gurjar (2011) for example, found that: "… irrespective of constant emissions over each month, the estimated tracer concentrations are invariably 40% to 80% higher in the winter months (November, December, and January) and 10% to 60% lower in the summer months (May, June, and July), when compared to annual average for that year."*

* From the various comparisons made with conditions in China, it is evident that the authors are somewhat surprised by the severity of the pollution problem in North India. Yet the magnitude of the PM2.5 problem in North India is becoming quite well documented, especially in literature focused on remotely sensed PM2.5 levels. The evidence is now quite clear that India has overtaken China for population-weighted PM2.5 levels, while other large countries (Bangladesh, Pakistan, Nigeria, Egypt) can be even more polluted. In my view, the focus on China's pollution in the popular imagination – even when nowhere near the highest in the world – arises in part because of the rapid improvement in the availability of ground-based data there. See for example Shaddick et al, ES&T 2018:
https://pubs.acs.org/doi/pdf/10.1021/acs.est.8b02864

*Yes, during the writing of this paper, we were also more and more convinced of that. But China still has a worse reputation than India indeed, that is why we also try to emphasize the problem in India in our paper.*

* A discussion of data quality and measurement uncertainty for the in-situ CPCB pollution measurements would be appropriate. Do the surface measurements appear to be generally well calibrated and reliable?

*We could not get any information on the data quality unfortunately*